
# FATES: A Flexible Analysis Toolkit for the Exploration of Single Particle Mass Spectrometer Data

Camille M. Sultana[1], Gavin Cornwell[1], Paul Rodriguez[2], Kimberly A. Prather[1,3] *

[1]Department of Chemistry and Biochemistry, University of California, San Diego, La Jolla, 92093, USA
[2] San Diego Supercomputer Center, University of California, San Diego, La Jolla, 92093, USA
[3]Scripps Institution of Oceanography, University of California, San Diego, La Jolla, 92093, USA

*Correspondence to:* Kimberly A. Prather (kprather@ucsd.edu)

**Abstract.** Single particle mass spectrometer (SPMS) analysis of aerosols has become increasingly popular since its invention in the 1990s. Today many iterations of commercial and lab-built SPMS are in use worldwide. However supporting analysis toolkits for these powerful instruments are either outdated, have limited functionality, or are versions that are not available to the scientific community at large. In an effort to advance this field and allow better communication and collaboration between scientists we have developed FATES (Flexible Analysis Toolkit for the Exploration of SPMS data), a MATLAB toolkit easily extensible to an array of SPMS designs and data formats. FATES was developed to minimize the computational demands of working with large datasets while still allowing easy maintenance, modification, and utilization by novice programmers. FATES permits scientists to explore, without constraint, complex SPMS data with simple scripts in a language popular for scientific numerical analysis. In addition FATES contains an array of data visualization GUIs which can aid both novice and expert users in calibration of raw data, exploration of the dependence of mass spectra characteristics on size, time, and peak intensity, as well investigations of clustered data sets.

## 1 Introduction

Single particle mass spectrometers (SPMSs) yield the size and chemical composition of individual aerosol particles in real-time. SPMSs can generate tens of single particle mass spectra per second utilizing laser desorption-ionization (LDI). However mass spectra generated by LDI can exhibit large particle-to-particle variation (Gross et al., 2000; Wenzel and Prather 2004; Zelenyuk et al., 2008). Thus SPMSs generate both large and highly complex datasets, requiring sophisticated data analysis techniques for exploration and distillation of information.

As Table 1 illustrates, individual laboratories have independently developed a variety of SPMSs, and two commercial versions have also been produced. Due to the many iterations of SPMSs that exist and the lack of a standard data format, individual laboratories have had to build their own data analysis software though these toolkits are often not reported in the literature (Table 1). Only two of these data analysis toolkits have been made publicly available, YAADA (www.yaada.org) and ENCHILADA (www.cs.carleton.edu/enchilada), both of which are specific to the aerosol time-of-flight mass spectrometer

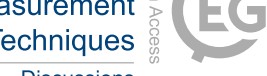

(ATOFMS), a version of SPMS.  Despite their age these toolkits are still utilized, with YAADA being the toolkit of choice for the burgeoning SPMS community in China.  The differences and limitations between these two software tools have been extensively described previously (Gross et al., 2010) but a brief summary is given here.  YAADA is an object-oriented framework implemented in MATLAB that allows user-developed script-based data exploration and can also leverage the extensive set of built-in functions

within MATLAB.   This allows a degree of flexibility in creating graphical outputs and exploring ATOFMS data in tandem with other data types. However, the extensive amount of code required at the time of development to create the object-oriented framework for YAADA has made the toolkit highly susceptible to updates and changes in MATLAB.  Thus continued use of YAADA either requires using outdated MATLAB versions or extensive maintenance of the scripts underlying the toolkit. Also

considerable knowledge of YAADA-specific data classes and framework in addition to general MATLAB understanding is required to be able to manipulate the data. Additionally, YAADA's accessibility is limited for novice users as there are no (graphic user interfaces) GUIs for data exploration.  In comparison Enchilada is a software package with a graphical user interface.  Therefore data analysis functions and workflows built into Enchilada are leveraged by interacting with the GUI, without the need to create scripts

or interact in a command line interface.  However any addition of functionality requires modifying the underlying source code and rebuilding the software.  Enchilada relies primarily on SQL for accessing and storing the mass spectra database, and Java for implementation of the GUI, though a number of other drivers, toolkits, and C++ are also integrated into its implementation.  Thus modifications are a significant programming task and likely infeasible for scientists not highly experienced in programming and computer

science.
     Motivated by the continued use of SPMS and the limitations of the currently available software we have developed a new flexible analysis toolkit for the exploration of single particle mass spectrometer data (FATES).  To encourage the widespread adoption of this toolkit it was purposely designed in an extensible manner to adapt to the ever evolving and varied implementations of SPMS.  It is clear that building open

source tools in a standard, well known platform, and creating a work flow with user defined parameters for data analysis would be beneficial to the SPMS community, increasing the rate of knowledge discovery and enabling collaboration between researchers.  For example, maintenance and alterations of the software should be easily accessible to chemists and aerosol scientists without extensive training in computer science.  In addition, any new toolkit should not be explicitly limited to expected common analyses, which

may be built into GUIs, but should give the user complete freedom to access, explore, and utilize SPMS data and also integrate with other temporally and spatially resolved data sets.  Finally any framework needs to make careful consideration of both memory and speed constraints imposed by the possible large size of SPMS data sets.  Given these constraints the FATES toolkit was developed completely in the MATLAB environment.  MATLAB is a popular language for numerical data analysis by scientists because it has an

extensive library of well-documented built-in functions, utilizes libraries optimized for speed in matrix manipulation, and can support both graphical and script based exploration of data.  By taking advantage of





native MATLAB data types, FATES is easier to maintain and computationally more efficient than YAADA, the previous publicly available MATLAB toolkit for SPMS analysis. The FATES framework allows users to creatively explore their data without previous assumptions or constraints with simple scripts and by

leveraging built-in MATLAB functions. Additionally FATES offers a suite of GUIs for interactive visualizations which can aid both novice and expert users in calibration of raw data, exploration of data sets using temporal, size, and mass spectral filters, as well as investigations of clustered data sets. FATES is the first publicly available SPMS toolkit to allow creative, efficient, script-based data mining along with GUI based visual data exploration and calibration all within a single programming environment.

## 2 FATES software description

FATES is implemented completely in MATLAB. No other languages, drivers, or software are needed to utilize FATES. In addition FATES was purposely developed in a manner that demands few presumptions about the instrument, particle, and spectra variables collected by the SPMS. For example one SPMS may only record the speed and time of detection for each particle while another SPMS may also record the

power of the desorption/ionization laser pulse. These differences are handled easily as FATES allows users to specify, define, and change the instrument, particle, and spectra variables they would like imported into and saved to a study. To make these alterations users only need modify simple scripts where the desired variables are listed and then these changes are carried over throughout the entirety of the source code. This flexible but simple design gives high utility for the SPMS community because it prevents users from

needing expert knowledge of any language and having to search for and make line-by-line or structural changes within the source code. Detailed instructions for making these simple modifications are included in the FATES manual and commented within the code. In addition FATES avoids the explicit creation of new class objects, which minimizes the lines of source code and number of scripts by over an order of magnitude when compared to YAADA. This greatly minimizes the maintenance needed to keep FATES

compatible with future versions of MATLAB. FATES has been tested for compatibility with MATLAB versions 2014b through 2015b.

### 2.1 FATES data architecture

SPMS data imported within FATES is stored within separate variables for the experiment description, the particle data, and the spectra data. A SPMS dataset imported into MATLAB via FATES is referred to as a

FATES study. Each FATES study stores a data structure that contains a number of user-defined fields (e.g. Instrument Name, Operator, Location) to describe the experiment in which the data within the study was collected. Each row of the structure describes a unique experiment, which pertains to a unique experiment id. All particle data (e.g. speed, power of desorption/ionization laser pulse) is stored in a MATLAB matrix. Logically, the data mostly consists of 1-many-relationships from study to experiment, experiment to

particle, and particle to spectral peaks. There is also a polarity identified with each spectra. The data is most typically loaded once then is often accessed and filtered in bulk. Therefore, it is more efficient to





organize the observed measurements into denormalized matrices for particle and spectral data, where key information is duplicated in each matrix.

More specifically, each particle within a FATES study has a unique two-column particle id. The first column of the particle id is the experiment id, previously described, to which the particle belongs. This framework allows users to easily select for particle or spectra data collected during a specific experiment within a FATES study that contains data from multiple experiments.

The mass spectral data for all particles in the FATES study is held in an external binary file. Users can easily and quickly retrieve spectra peak data (e.g. m/z, area, height) for user-selected particles using functions provided by the FATES toolkit. The spectra data when imported is then stored in MATLAB cell arrays or matrices. Each peak for all the spectra within a FATES study has a unique four-column peak id. The first three columns of the peak id are the experiment id, particle id, and the polarity indicator of the spectra to which the peak belongs. Note that each FATES study contains auxillary data structures that list the name of the variable (e.g. particle speed, peak area, peak id) that each column in a data matrix holds. Thus all data within a FATES study is self-contained and self-described. Therefore despite the flexibility of the FATES framework, users can still share FATES studies without confusion or need for external READ ME files to determine the source and identify of the data.

**2.2 FATES optimization**

Considerable work has been completed to optimize the FATES framework for memory demands, speed, and ease of use. An ATOFMS dataset collected at Bodega Bay, CA in February and March of 2016 is used throughout this paper to illustrate the speed of data analysis within the FATES toolkit. This dataset contains 1,386,042 dual-polarity single particle mass spectra as well as particle data for an additional 11,454,356 particles that were detected in the light scattering region but did not generate spectra. All FATES analysis is performed in MATLAB 2014b with an Intel Core i7-4930K CPU running at 3.4 GHz with 16.0 GB of RAM.

To begin working with a SPMS dataset a new FATES study has to be created. This process only needs to occur once for any dataset, but the source-code was still designed to minimize the time for study initialization. Despite the large size of the Bodega Bay dataset, the creation of the FATES study only took 28.4 minutes. Speed comparisons are made using the same computer utilizing a version of YAADA, which had been maintained by Kim Prather's research group to be compatible with MATLAB 2013a. Even initiating a subset of the Bodega Bay study roughly a tenth of the FATES study (127,077 dual-polarity mass spectra) in YAADA still required 20.8 minutes. FATES has also been designed so that additional data can be added to an existing study without having to reinitialize the entire data set. This is especially useful for field studies, where daily examination of the data is required, but initialization of increasingly large data sets can become onerous and time consuming.

Once a FATES study is initiated, it is crucial to efficiently handle the spectra data. Users may desire to examine datasets with millions of mass spectra and each spectrum can contain hundreds of peaks. SPMS



spectra data formats usually contain mass-to-charge (m/z) and area for each peak, but may also specify peak width, peak height, and other values. This amounts to many gigabytes of data, and therefore the trade off between making all the spectral data available and managing memory requirements had to be taken into consideration. MATLAB facilities for tables were considered, but they are more appropriate for heterogeneous data, whereas in our case all the spectra data is numeric or binary indicators. We also found MATLAB memory mapped files to have unpredictable performance, and it was difficult to append data rows because matrices are stored in column order. We determined the best way to build up and maintain a large matrix of spectra data, without keeping it in memory, was to create a single external binary file, append to it as needed, and provide a lightweight interface so that FATES programs, or other users, could easily execute functions against the file. Essentially, this interface is an API (application programming interface), which takes a regular MATLAB command or script, shuffles data in/out of memory in blocks of rows, executes the commands against the data in memory, and gathers results. The block sizes are set to default values that are reasonable for current workstation capacities, but can also be changed as appropriate in the future. The possible commands are unconstrained, but summaries and filtering operations are most appropriate and most likely to be called for.

In addition, the binary format minimizes both the time required to write and retrieve spectra data as well as the storage requirements for the file. Retrieving all 1,386,042 dual-polarity mass spectra in a single call from the external binary file created for the Bodega Bay study and loading it into a MATLAB array only took 3.3 minutes. It is important to note that this example is used for benchmarking purposes, but rarely would users need or choose to load into and hold all spectra information for entire large datasets within memory at the same time. The FATES framework automatically employs data pointers so that the whole binary file does not need to be read if the user is only attempting to retrieve spectra from particles which make up a subset of all the data in the FATES study. Retrieving a subset of 400,000 mass spectra that are contiguous in location in the binary file (i.e. the raw data files from which the study was created were contiguous) only required 26 seconds, a subset of 50,000 contiguous spectra required 2.7 seconds, and a single spectrum 0.12 seconds. In comparison, retrieving all 127,077 dual-polarity mass spectra from the smaller subset study using the YAADA toolkit took 42.5 seconds and a subset of 50,000 contiguous spectra required 17.3 seconds. Searching and sorting data by particle information is also quickly performed in the FATES framework. By holding all hit particle data in memory, any operation querying the particle data does not require any data input/output calls and therefore is nearly instantaneous in MATLAB. For example retrieving the particle ids for all sub-micron particles from the 1,386,042 hit particles in the Bodega Bay study only took 0.01 seconds, while performing a similar analysis in the 127,077 particle YAADA study required 0.6 seconds.

The quickness of the FATES framework depends partially upon minimizing retrieval calls to external files outside of the MATLAB workspace. Thus formatting of the data held within the MATLAB workspace has been carefully considered to minimize the memory demands of the FATES framework. Because spectra data is held in an external binary file users can choose to store spectra data in the study at a high resolution



without increasing the study's working memory. When retrieving spectra from the external binary file users may specify the resolution to hold the data in the workspace. This feature allows users to tailor the resolution of the spectra in the workspace to its application and therefore the memory requirements. Mass spectra data loaded into the MATLAB workspace is stored in a single-precision floating-point format, saving memory compared to the standard MATLAB double-precision format, which requires twice the

space. Particle data stored within a FATES study has also been formatted to minimize memory demands. If the user loads data into a FATES study for both detected particles that generated mass spectra (hit) and detected particles that did not generate spectra (missed) only hit particle data is stored in the particle matrices in MATLAB. Most data analyses utilize spectra and therefore only hit particle information is necessary, but hit particles usually make up a small fraction of total particles detected by the light scattering

region of the SPMS. Therefore storing missed particle data in MATLAB memory would take up large amounts of space needlessly. All missed particle data is written to an external binary file and can be loaded by the user into MATLAB using a script provided in the FATES toolkit. Furthermore particle data stored in MATLAB memory is split between a single-precision and double-precision matrix. It is not necessary to store most data collected for particles (e.g. speed, laser power) in a double-precision format, so this choice

further relieves the space required to store all particle data in memory. Therefore storing data for 1 million hit particles in memory where three variables require double-precision format (particle id, time) and three variables only need single-precision format (speed, size, laser power) only requires 0.036 GB, which is very feasible for most modern desktop computers. Finally because all SPMS data when loaded into a FATES study is held in native MATLAB data types, interacting with the data requires very few FATES

specific functions. Almost all common analyses can be patterned off a basic script relying on a handful of MATLAB built-in functions and matrix indexing, making the FATES framework accessible and powerful for both expert and novice users.

**3 Data analysis within FATES**

In this section we provide a brief overview of common analyses that can be performed on SPMS data

within a FATES study. However it should be noted that it is impossible to describe or predict all data analyses and plotting options easily available to FATES users due to the extensive library of built-in and user-developed MATLAB functions. A large array of analyses can be performed using concise code. Utilizing logical indexing, particles and spectra can be filtered using any single or combination of particle and mass spectra characteristics (e.g. particle size, peak area at a certain m/z, etc.). Binning of particles and

spectra by these characteristics, such as binning data based on time, can be accomplished in a single line with the built-in function *histc*. Additionally lists of particles can be compared with the built-in function *intersect*. Grouping data based on algorithmic clustering of the spectra is also easily performed. Clustering methods commonly used by the SPMS community such as k-means, hierarchical clustering, and k-medoids are built-in to MATLAB and the ART-2a algorithm, popular among ATOFMS users, is supplied in the

FATES toolkit. Clustering data, which necessitates a large number of matrix operations, can be performed





quickly even with naïve user scripts because MATLAB utilizes BLAS, LAPACK and proprietary libraries which speed-up common linear algebra computations. Clustering 100,000 particles from the Bodega Bay study with ART-2a (vigilance factor = 0.80, learning rate = 0.05) in the YAADA study required 70 minutes, however improvements in the ART-2a scripts in FATES allows the same analysis to be completed in only

2.1 minutes. Using the built-in MATLAB k-means function the same data was grouped into 15 clusters in 2.9 minutes (77 iterations) in FATES. Finally other types of data can be easily loaded into MATLAB and examined along with the SPMS data.

**4 Exploration of data utilizing FATES GUIs**

**4.1 guiFATES: spectra visualization, grouping, and exploration**

While the FATES toolkit allows flexibility in script-based SPMS data analysis, graphical tools can also be an effective way to explore the data and quickly identify trends and patterns. To this end the FATES toolkit includes GUIs, built within MATLAB, which allow users to easily examine trends in spectra based on particle size, time, and cluster and spectra characteristics. Figure 1 is a screen capture of the FATES spectra explorer guiFATES, displaying data for 46,432 particles. This spectra explorer has been modeled

after ClusterSculptor, a SPMS data analysis GUI developed by Zelenyuk et al. (2008) that has not been made publicly available. To initiate guiFATES the user provides the *guifates* function with the mass spectra, size, time, and cluster data for a set of particles. A description of the functionality and abilities of guiFATES is given below.

The main panel of the guiFATES display is the heat map of the individual particle mass spectra. Each row

is an individual mass spectrum with peak intensity indicated by color, with red being high intensity and blue is low. The user can select in the display parameters to choose to display the provided mass spectra peak intensity utilizing a linear or log10 scale. The logarithmic scale makes it easier to visually detect relatively small peak intensities in the spectra, while the linear scale helps users visualize absolute differences between peak intensities. In Fig. 1 the logarithmic scale has been selected. The size and time

of collection for the particles generating each spectrum is displayed in the plots on the left and far left respectively, while clustering information is displayed on the right. The cluster or group assigned to each particle is indicated by the color of the points on the right while the location on the x-axis is a user provided clustering statistic for each particle. The clustering statistic provided for display in Fig. 1 is the dot product of each normalized particle spectrum with the normalized representative spectrum of the cluster to which

the particle had been assigned. However, it is important to note that the user can provide any clustering or neighbor statistic they feel is effective for exploring their data set. The top plot in guiFATES is the average of all the provided spectra, and the plot immediately below that is a display for the average spectra grouped by cluster. The user selects which average cluster spectra to display in the display parameters. The line color in the average cluster spectra plot matches the colors used to indicate the assigned cluster for each

particle in the right most plot. The bottom of the guiFATES windows contains all the display, sorting, filtering, and grouping parameters that the user may select and change.





guiFATES provides the user with many options for displaying and exploring the data. A checkbox allows the user to display all data with or without grouping by cluster. In addition the user can select to sort the data by any of the particle metrics (size, time, or cluster statistic) or by a m/z value in the spectra. In Fig. 1

the data is displayed by cluster and sorted by size. Figure S1a is a screen capture where the same data is not grouped by cluster and has been sorted by peak intensity of m/z -35. While users may initially provide guiFATES with a large amount of data, they will likely desire to display smaller selections at a time to enable better visual exploration. This can be accomplished in a number of ways within guiFATES. Users can use mouse clicks to quickly zoom in and out of a single plot using MATLAB's native figure handling

capabilities. guiFATES is designed so that when this occurs all plot axes within the GUI are scaled appropriately and instantaneously. Figure S1b is a screen capture where the user utilized this functionality to select the bottom half of the particles in Fig. S1a and also decreased the range of the m/z values displayed. For more complex selections users can enter in filtering parameters so that displayed particles only fall within a desired range of size, time, clustering statistic value, peak intensity of a certain m/z value

or any combination thereof. Figure S1c is a screen capture where the data, sorted by cluster, has been filtered by size (1-2 um), m/z -35 peak area (0-3000), and clustering statistic (0.8-1). Finally users can also choose to only display select clusters. Figure S1d is a screen capture utilizing the same filters as in Fig. S1c however limiting the display to only clusters 2 and 5.

These visual sorting and filtering methods enable users to visually discover trends differences and

similarities within and between groups of particles. Due to the high variability in mass spectra generation inherent to single particle laser desorption-ionization techniques, clustering algorithms utilized to group SPMS mass spectra within a dataset often do not generate a one-to-one relationship between the number of chemical particle types in the population and spectra clusters generated. The necessity of expert knowledge to either combine multiple spectra clusters, generated algorithmically, into a single chemical particle type

or to further split clusters into smaller groups has been noted frequently within the SPMS community (Gross et al., 2000; Wenzel and Prather 2004; Zelenyuka et al., 2008; Zelenyukb et al., 2008; Giorio et al. 2012). guiFATES aids this process by allowing users to combine any two specified clusters or split any cluster during the data exploration process. Combining clusters is easily performed by selecting the desired clusters from a list. To split a cluster a user simply selects the cluster to be split and then clicks on the

display plot where the split should be performed. Users can choose to output the particle identifiers of any cluster currently in the guiFATES window to the MATLAB workspace. The advantages and benefits of this general method of data visualization and exploration for refining particle clusters has been extensively discussed (Zelenyuk et al. 2008) and with the publication of FATES will be available to the SPMS community at large. All plotting, sorting, filtering, and grouping applications of guiFATES have been

tested on a set of 100,000 particles with dual-polarity mass spectra, and at this size all updates to the displayed plots occurred nearly instantaneously, making guiFATES an appropriate and efficient tool for the large data sets common to SPMS analysis.

**4.2 dendroFATES: hierarchical cluster relations**





FATES also includes two supplementary GUIs which allows the users to graphically select the particles to
feed into the guiFATES spectra explorer. dendroFATES is a GUI where the user supplies the clusters and
representative cluster mass spectra output from any clustering algorithm of the user's choice. The clusters
are then automatically grouped into a cluster tree by a hierarchical analysis performed within MATLAB
which is displayed in the dendroFATES GUI window. Hierarchical analyses have been utilized previously
with SPMS data sets, but a brief description is given here. The dendrogram links clusters in a binary
fashion creating new groups which are then further linked. The height of each linkage indicates the
calculated distance between the group in the left and right branch of the linkage. Lower linkage heights
indicate a higher degree of similarity between groups and large distances between levels in the dendrogram
are indicative of natural divisions in the dataset. Figure 2 is a screenshot of the dendroFATES window
with a dendrogram generated from the thirty most populous clusters generated using the ART-2a algorithm
to cluster a subset of 166,666 particles from the Bodega Bay dataset. Zooming in and out of the
dendrogram is handled by MATLAB's native graphics functionality makes it possible to supply
dendroFATES with hundreds of clusters and still explore the cluster tree quickly and intuitively. Because
the dendrogram allows the user to easily visualize similarities and natural groupings of clusters generated,
it is an excellent tool to select clusters for further exploration of the particle and spectral data using the
guiFATES tool. Clicking linkages in dendroFATES automatically opens a guiFATES window displaying
all particles belonging to the selected node. The user can choose to output the particles to guiFATES with
their original cluster labels or group all clusters belonging to the left or right branch of the linkage together.
When a linkage is selected the fractional cluster contribution to the selected node is displayed on the right
in the dendroFATES window and the fraction of the selected node to the total population is also displayed
in text. Figure S2a illustrates the guiFATES window generated with the node selection made in Fig. 2
when the user chooses to group particles by their cluster label, while Fig. S2b is the same data grouped by
left or right branch. As illustrated in Fig. S2a, when guiFATES is populated by dendroFATES the clusters
are displayed in the same order as displayed in the dendogram. Therefore very similar clusters are adjacent
in the guiFATES window, assisting intuitive visual comparisons and combinations of data. Because all
FATES GUIs are in MATLAB and the user can also access the data programmatically, it is straightforward
and fast for the user to iteratively select clusters from the dendrogram in dendroFATES, refine them in
guiFATES, output new clusters to the workspace, and feed the new cluster results back into dendroFATES
until the user is satisfied with the grouping of the data set.

**4.3 scatterFATES: user defined particle relations**

The complexity of SPMS datasets means there are numerous relationships that could be explored, and
predicting all desired comparisons is impossible. scatterFATES is another GUI used to populate
guiFATES with user selected particles. However, rather than grouping particles via clusters as in
dendroFATES, scatterFATES creates a scatter plot of particles using any two particle data metrics the user
supplies as the axes. The points are then color coded by cluster or group. Figure 3 is an example



scatterFATES window, where the -35 to -93 m/z ratio is plotted against particle size for the 166,666 particles that had been previously clustered. Once a scatter plot is created in scatterFATES the user can click on the figure to draw regions within the scatter plot as shown in Fig. 1. All particle data within a created region can then be selected and automatically populated into guiFATES for spectra visualization and exploration.

### 4.4 calibFATES: raw spectra calibration

FATES has been designed so that all aspects and functionalities of SPMS data analysis and exploration are contained within a single programming environment and language. To this end we developed calibFATES, a GUI to quickly scan through raw spectra data files before importation into FATES and generate calibrations to convert raw time-of-flight spectra to mass-to-charge spectra. calibFATES is currently written to be able to read the raw spectra files generated by the ATOFMS and TSI ATOFMS, however could be easily modified to read in any raw spectra file. Figure 4 is a screenshot of a calibFATES window displaying a single uncalibrated raw spectrum. Users can use arrow keys to scan through and display spectra contained in any raw spectra files within the folder. A calibration can be generated by setting selected times to entered m/z values. Those times can be chosen either by clicking on the plot or manually entering the times into a text box. Calibration parameters can be output to a text file for future reference and any calibration file generated can be loaded into and applied to the raw spectra in calibFATES so that the spectra are displayed as calibrated mass spectra rather than time-of-flight spectra. calibFATES allows SPMS users to quickly visually examine generated spectra on the fly without any time consuming processing, even during data acquisition, to ensure the quality and consistency of the data being acquired.

### 5 Conclusions

FATES is the first software package for SPMS data sets to include flexible script-based data analysis and graphical user interfaces for data exploration integrated within a single programming language. Because FATES is designed to be easily extensible to diverse input data formats and implemented completely in MATLAB, a highly documented language popular among scientists, it should be accessible and employable across the SPMS community despite the many independent instrumental designs. SPMS data importation and programmatic and graphical data analyses can be performed quickly in FATES even for large datasets thanks to both speed and memory optimizations and utilization of native MATLAB data types and built-in functions. Within a FATES study data is structured so that complex analyses can be performed using concise code with little reliance on FATES specific functions. In addition a set of GUIs with many display, sorting, filtering, and grouping functionalities have been developed to assist both expert and novice users to intuitively visualize a complex SPMS dataset and create robust particle groupings. For these reasons we believe FATES will greatly improve the efficiency of data processing and knowledge discovery from SPMS datasets.

### 6 Code Availability



The FATES software package, an extensive manual, and an example dataset is available at github.com/CMSultana/ FATESmatlabToolKit. This site is a forum where updates to the code and new functions can be shared amongst the SPMS community.

*Competing Interests*. The authors declare that they have no conflict of interest.

*Acknowledgements*. This work was funded by the National Science Foundation through the Center for
Aerosol Impacts on Climate and the Environment (CHE 1305427). Any opinions, findings, and conclusions or recommendations expressed in this material are those of the authors and do not necessarily reflect the views of the National Science Foundation.



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





| SPMS version | Analysis Toolkit Utilized |
|---|---|
| *Lab developed Instruments* | |
| ALABAMA[a] | CRISP (IGOR toolkit)[b] |
| ATOFMS[c] (UF-ATOFMS[d]) | YAADA (MATLAB toolkit)[e] |
| PALMS[f] | Not reported |
| RSMS[g] (RSMS-II[h], RSMS-III[i]) | Not reported |
| SPLAT[j] (SPLAT II[k], mini-SPLAT[l]) | SpectraMiner[m], ClusterSculptor[n] |
| *Commercial Instruments* | |
| Guangzhou-Hexin ATOFMS/ SPAMS (currently manufactured)[o] | YAADA |
| TSI ATOFMS (discontinued)[p] | YAADA, ENCHILADA[q] |

[a]Brands et al. 2011; [b]Klimach 2012; [c]Gard et al. 1997; [d]Su et al. 2004; [e]Allen 2005; [f]Thomson et al. 2000; [g]Carson et al. 1995; [h]Phares et al. 2002; [i]Lake et al. 2003; [j]Zelenyuk and Imre 2005; [k]Zelenyuk et al. 2009; [l]Zelenyuk et al. 2015; [m]Zelenyuk et al. 2006; [n]Zelenyuk et al. 2008; [o]www.tofms.net/content.aspx?info_lb=387&flag=103; [p]www.tsi.com/aerosol-time-of-flight-mass-spectrometers-series-3800; [q]Gross et al. 2010


**Table 1: Summary of SPMSs developed and data analysis packages used**





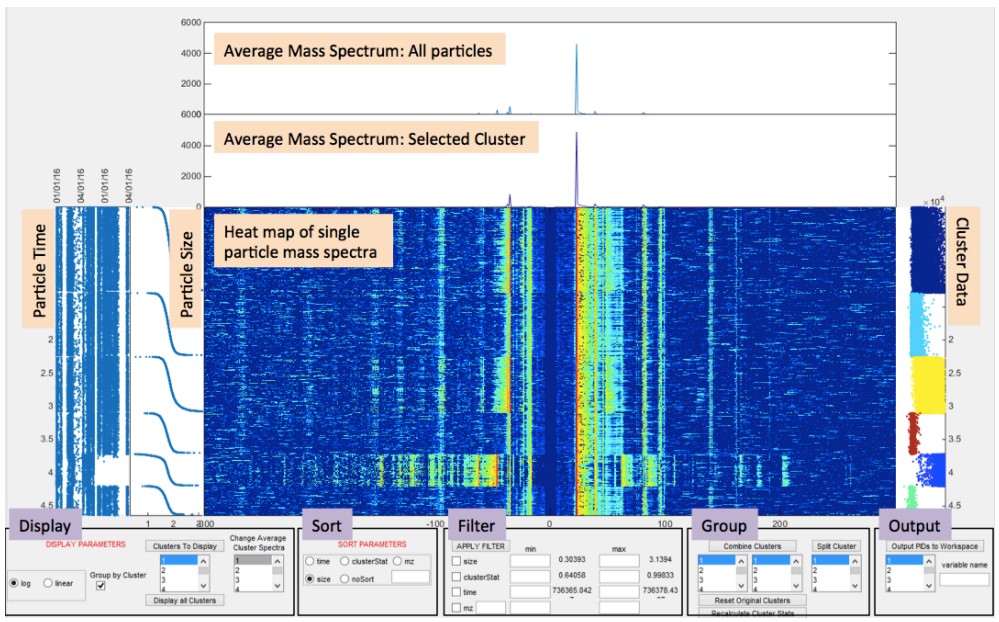

Figure 1: Screen capture of a guiFATES window with data from 46,432 individual particles.






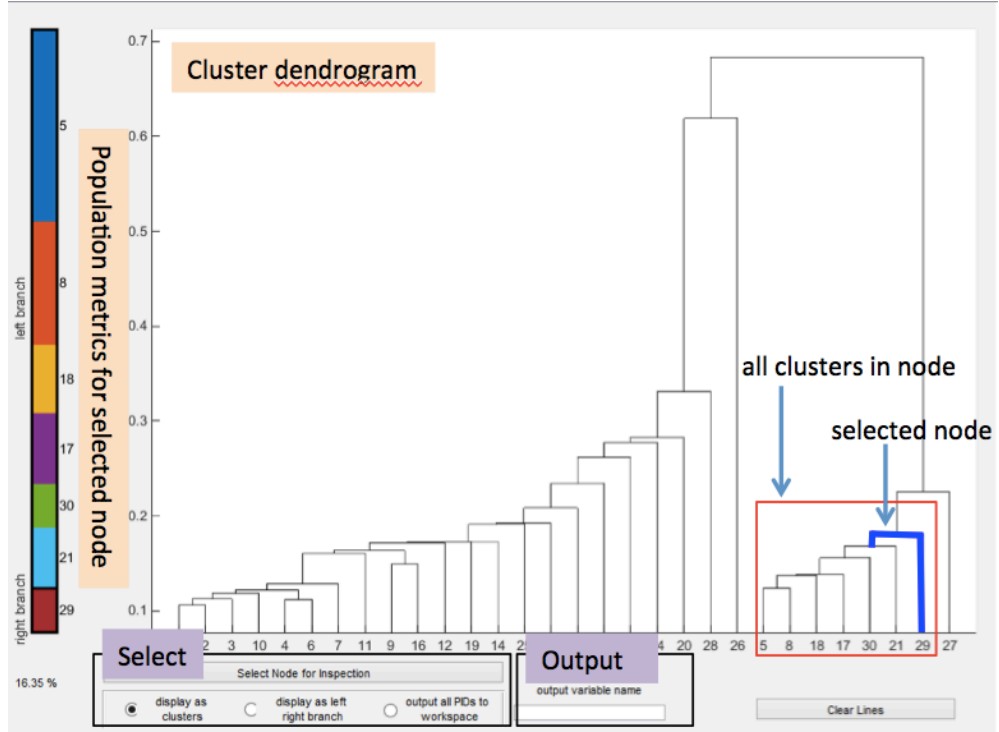

**Figure 2: Screen capture of a dendroFATES window showing the cluster tree or dendrogram for 30 input clusters. The cluster contributions to the user selected node are shown in the plot on the left. The particle data for the selected node are automatically plotted in a guiFATES window (Figure S1).**






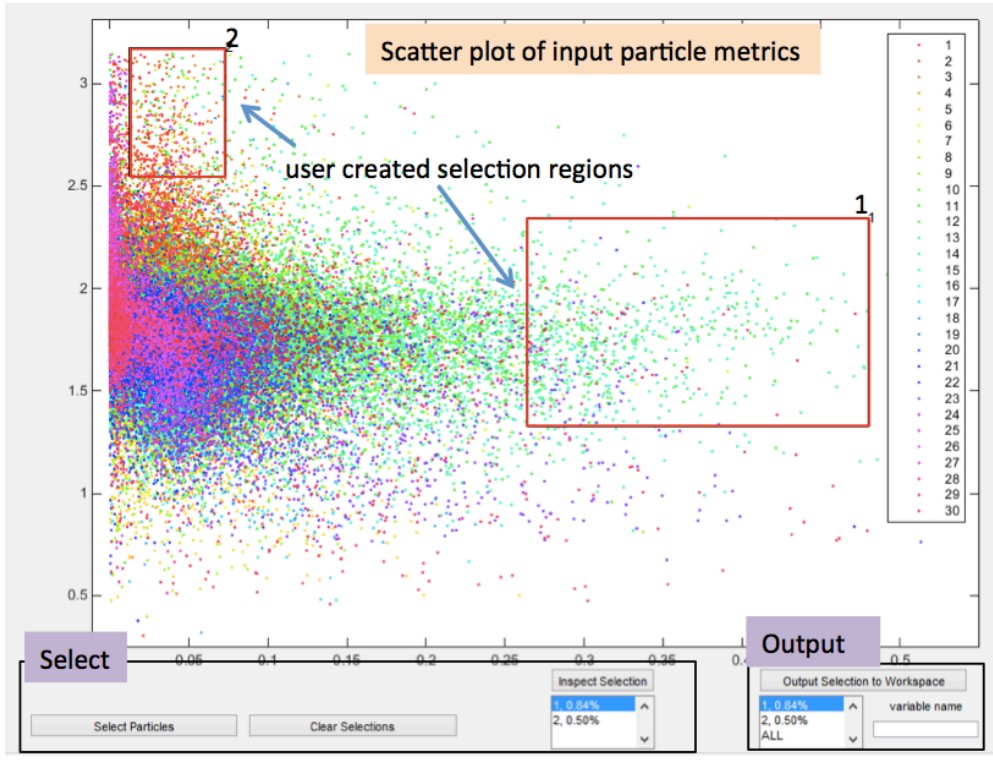

Figure 3: Screen capture of a scatterFATES window showing the -35 to -93 m/z ratio plotted against particle size for 166,666 particles. Any two particle metrics can be input into scatterFATES. Two regions have also been created by the user for further inspection in guiFATES.






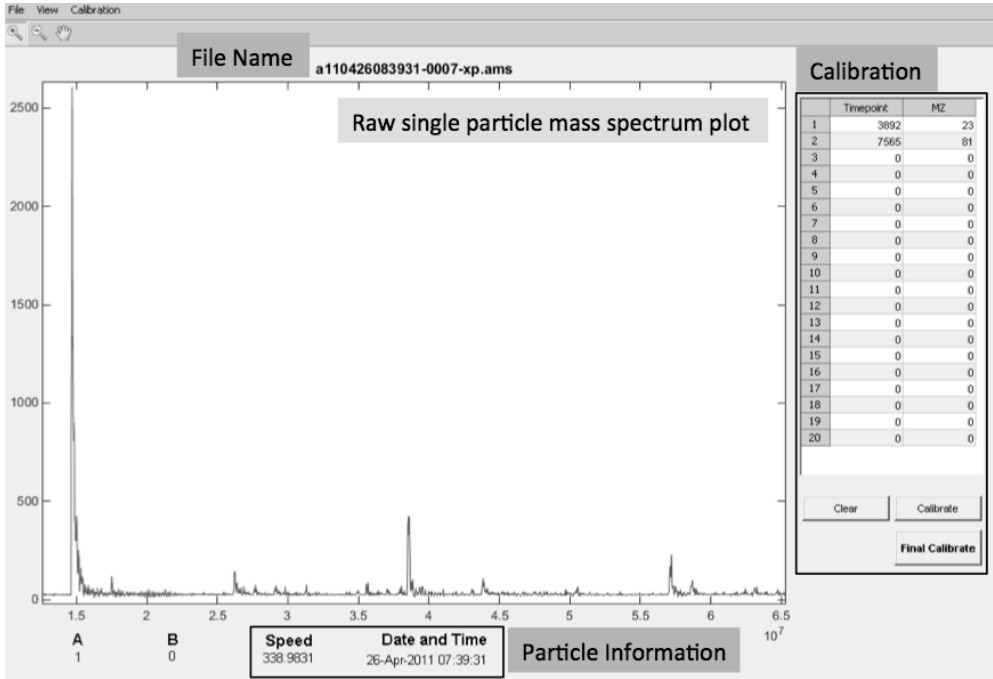

**Figure 4: Screenshot of a calibFATES window displaying a single particle uncalibrated mass spectrum.**
**Calibration data is input and displayed on the right and particle size and time are displayed on the bottom.**