# Peer review of "Figure S1. Screen captures of a guiFATES window with the same data as in Figure 1 filtered and sorted in various ways. a) Entire population sorted by m/z -35 peak intensity. b) Zoom in of S1a selecting the bottom half of the particles displayed and a decreased m/z range. c) The data, sorted by clust"

_Atmospheric Measurement Techniques, 2016_

## Referee Comment (RC1) · Anonymous Referee #1 · 13 Nov 2016

This is a great piece of coding and work. Owing to the great variety of data formats of particle mass spectra and the copious numbers and types of clustering and relational information that is sought, having the flexibility provided by FATES is critical. The authors have included every iteration of data analysis and presentation that I could think of. Nonetheless, I am confident that the user base for this wonderful analytical tool will grow quickly and with the added insight of many users will become even more useful.

I have two primary concerns however: (1) Operation has been demonstrated only with the authors' data format. I believe that, before publication, the authors should experiment with different data formats from the intended users of the program. I could find little discussion of how data entry for specific formats is or will be accomplished. Having

**[AMTD](AMTD)**

Interactive
comment

had a copy of the program to test would have been very beneficial. Unfortunately, the link in the manuscript is not working. (2) How will the authors overcome the potential limitation of becoming obsolete, or at least not readily upgradeable or adaptable, with future versions of Matlab, as is the case suffered by YAADA.

Again, overall I feel that the authors are providing a very important to harmonize particle mass spectrometer data across a user base of data formats that has developed quite haphazardly over the past decade or so. In my opinion though, I believe that publication at this point is precocious until the program has been tested to some extent by others.

——————————————————————

---

## Author Comment (AC1) · 16 Nov 2016

Response to comment 1: We are sorry that the link in the manuscript was not working for you to explore the toolkit. We believe that we have fixed the permissions on github so that it is now open to the public to download. Please let us know if you continue to have trouble. Along with the code we have provided a very comprehensive, 33 page, FATES manual. Over ten pages are dedicated to describing the raw data formats, data importation process, current FATES structure, and instructions for altering for adapting the structures and code for compatibility with different data formats. Due to the very detailed nature of this description we do not believe it is appropriate for discussion in the paper, though of course the information is important to provide to users. In

addition we will work on acquiring raw data from an outside SPMS group to validate the adaptability of FATES to different data formats.

Response to comment 2: As mentioned briefly in the paper YAADA was particularly susceptible to updates to MATLAB and difficult to maintain because the YAADA framework created and then utilized data types which were not native to MATLAB. Because YAADA constructed 6 new data classes YAADA couldn't automatically leverage many of the built-in MATLAB data handling functions. Numerous scripts had to be included in YAADA to perform very simple operations mirroring the capabilities of functions already in MATLAB. Because FATES utilizes native MATLAB data types the code base required to operate the FATES toolkit is decreased by over an order of magnitude from YAADA. In addition, in FATES we have attempted to leverage matrix operations and low-level operations whenever applicable which are unlikely to be affected by foreseeable MATLAB updates. While any toolkit is going to require maintenance eventually as MATLAB updates, we believe that due to the relatively small and simple code base maintenance should not only be minimal but it would likely be straightforward to take advantage of future MATLAB enhancements. We also believe that by offering FATES on a platform built to promote open-source software development (github) it will be much easier for the community to collaborate and improve FATES.

---

## Referee Comment (RC2) · Anonymous Referee #2 · 16 Dec 2016

This manuscript provides a description of a flexible analysis toolkit for single particle mass spectrometer (SPMS) data, based in MATLAB, which represents an important and substantial contribution to the SPMS community. The authors have clearly put much thought and work into making the FATES toolkit as adaptable, user-friendly and maintainable as possible. Making the code publicly available is an important step towards transparent and reproducible data analysis. I have no doubt that different implementations and functionality in FATES will grow as it is adopted by the community.

General Comments:

I have two main concerns about this manuscript.

[Figure]

(1) The use of FATES has been demonstrated only with SPMS data from the author's research group and with only one type of SPMS. For this work to be published the authors must demonstrate clearly the successful implementation of FATES with very different types of SPMS data sets. Related to this point, the authors provide little information on data import and structure for different SPMS data formats. I understand that this information is available in the FATES manual provided on the GitHub page; however, the authors should consider including the manual as a supplement to this paper. The benefits of this (while being, admittedly, somewhat information heavy) will be more specific reference to information contained in the manual in the main text of the paper, and the ability for user's to cite the manual for justification of data analysis choices.

(2) While providing a valuable resource for the SPMS community, this manuscript should also seek to establish some central guidelines or standards for SPMS data analysis and interpretation. The authors indicate the necessity of expert knowledge in determining finalized chemical particle types from a dataset, and provide interfaces to facilitate these decisions. To help ensure the proper and educated use of these tools the authors should provide some concrete criteria by which the identification of chemical particle types should be made in FATES (e.g., criteria for evaluating linkage heights in dendroFATES, guidelines on evaluating cluster similarity using the dot product or other methods, discussion of advantages and disadvantages of different clustering approaches for SPMS data and criteria by which to choose a particular clustering approach). While this information is likely available in pieces in a variety of publications, a discussion of best-practices would really strengthen this work.

Specific Comments:

In general, this manuscript could be written in a more clear and concise manner. Intermingled with a description of the software seems to be sections that read more like an instruction manual. This paper could benefit from more consistency in the way the information is presented.

L95: Do the authors intend to test the compatibility of FATES with MATLAB 2016?

L109 (and other instances): "...a unique, two-column, particle identification (ID)."

Section 2.2: Much of the text in this section describes comparison of run-times for FATES and YAADA. This discussion is a bit difficult to follow and it might be easier if the run-time information was summarized in a table. When another type of SPMS data is included in this paper, similar information could be included provided the data can be analyzed in YAADA.

L318: Related to comment (2), above, how can it be objectively determined that the user should be "satisfied" by the FATES output? At present this seems incredibly subjective, and should be delineated.

Section 4.4: Are there standard procedures for mass-calibrating SPMS data? What is the smallest allowable number of peaks to be used in mass calibration?

Figures: In general, the figures are quite difficult to read.

---

## Author Comment (AC2) · 13 Feb 2017

Response to Comment 1: To keep the paper clear and concise, specific examples and instructions for changing the data formats are not detailed in the paper. However, an extensive FATES manual has been developed and is now included as a supplement to the paper. Where appropriate, references to specific sections of the manual are made throughout the entirety of the paper to guide the intended users of the program.

Also FATES has now been successfully implemented with three distinct SPMS datasets: ATOFMS (operated in Dr. Kim Prather's lab at the University of California, San Diego), ALABAMA (operated in Dr. Johannes Schneider's lab at the Max Planck Institute for Chemistry), and commercial TSI-ATOFMS (operated in a number of laboratories world-wide). Data files for these SPMSs have completely different organizations and structures and store a variety of variables that are not all shared between the datasets. However, the inconsistencies in the variables recorded within each dataset provided no barrier to implementation within FATES due to the toolkits designed flexibility.

Response to Comment 2: Unfortunately due to both the qualitative and inconsistent nature of SPMS spectra, developing "standard" criteria for creating robust particle types is well beyond the scope of this paper and is realistically not feasible across an unbounded set of particle types and a range of different instruments. First, it is important to note that the generation of single particle mass spectra will be heavily influenced not only by particle composition but also by SPMS instrumental design and operation. Different desorption and ionization laser wavelengths and powers, single-step versus two-step ionization methods, single polarity versus dual polarity acquisition, and the pre-processing functions for baseline determination, noise reduction, and peak picking algorithms will all influence the collected data. Furthermore, discussions on accurate methods to identify particle types not only span a range of mathematical and grouping techniques (Giorio et al., 2012; Gross et al., 2010; Murphy et al., 2003; Rebotier and Prather, 2007; Zelenyuk et al., 2008a), but also are usually specific to a narrow set of particles or even a single type (Pratt and Prather, 2009; Silva et al., 1999; Silva and Prather, 2000; Zawadowicz et al., 2016), and may also be instrument dependent (Hinz et al., 2006). Because of this setting, some sort of standard clustering threshold (whether using ART-2a, K-means, or a clustering algorithm of choice) will almost certainly leave some particle types split across multiple clusters and some clusters composed of multiple particle types (Murphy et al., 2003; Phares et al., 2001; Rebotier and Prather, 2007; Wenzel and Prather, 2004; Zelenyuk et al., 2008a). This reality means SPMS analysis on external aerosol populations will almost always require some form of expert knowledge and further analysis beyond initial algorithmic grouping, as specified in almost all field study publications (e.g. Dall'Osto and Harrison, 2006; Pratt and Prather, 2009; Qin et al., 2012). The various visual interfaces provided in FATES

were built to facilitate the flexibility needed to properly create and shape particle types and allow to more thoroughly and freely investigate both ungrouped datasets and initial groupings generated by various algorithms. Below we detail some of the specific hurdles to generating these specific recommendations from the reviewer.

As mentioned in the paper, it has been frequently demonstrated that the mass spectra generated via laser desorption/ionization (LDI) show wide particle to particle variation even for particles of identical chemical composition (e.g. Gross et al., 2000; Wenzel and Prather, 2004; Zelenyuk et al., 2008). This variation is exacerbated by variations in particle size (Reinard and Johnston, 2008; Zelenyuk et al., 2008a), inconsistency in laser fluence experienced by the particles (Steele et al., 2005; Wenzel and Prather, 2004), or spatial chemical heterogeneity within particles (Cahill et al., 2015; Cai et al., 2006; Zelenyuk et al., 2008b). Gross et al. (2000) show that for 1 um particles composed of 2,4- dihydroxybenzoic acid and ionized by an inhomogeneous laser, ART-2a generated 15-20 clusters even for very low vigilance factors (0.2-0.6), well below the threshold for initial clustering which is usually utilized for field study datasets ($\sim$0.7-0.8) where there is a need to separate spectra from actually distinct particle types (e.g. Dall'Osto and Harrison, 2006; Pratt and Prather, 2009; Qin and Prather, 2006). These problems in spectra variation are intensified for particle types where all particles are of similar composition, but within each particle there is spatial heterogeneity in the chemical makeup, such as "core-shell" morphologies. For example, Zelenyuk et al. (2008b) showed that 146 nm NaCl particles, coated in a 59 nm shell of dioctyl phthalate (DOP) can generate mass spectra with 100% contribution from DOP ions to 85% contribution from NaCl ions at a constant laser power of 0.38 J/cm2. Algorithmic analysis of these mass spectra would divide these results into an array of distinct particle types, despite the existence of only a single particle population. A wide array of particle types can exhibit chemical spatial heterogeneity, such as effloresced sea spray aerosols and particles which have undergone secondary processing (e.g. reaction with nitric acid, condensation of gas phase organic species, etc.). We have detailed just a few of the many examples in the SPMS literature, where particles with identical chemical compo-
sition generate widely disparate mass spectra.

Analysis of SPMS datasets is also made more difficult by the fact that a combination of matrix effects (Hinz and Spengler, 2007; Nash et al., 2006; Sullivan and Prather, 2005) and differences in ionization efficiency (Bhave et al., 2002; Ge et al., 1998; Gross et al., 2000; Hinz and Spengler, 2007; Spencer and Prather, 2006) result in non-quantitative mass spectra. As such, mass spectra can have large relative contributions from minor components of the particle composition and major components of a particle may only contribute negligibly to the mass spectra. This can result in distinct particle types having very similar mass spectra (Murphy et al., 2003; Silva et al., 1999; Zawadowicz et al., 2016). For example, one area of ongoing discussion is the accurate identification and discrimination between aerosolized dust and cells, because both types often share major ion markers (potassium, phosphate, organic nitrogen) with only frequently much smaller ion markers, such as silicates and aluminum, distinguishing between the two (Ault et al., 2011; Creamean et al., 2013, 2014; Holecek et al., 2007; Pratt et al., 2009; Zawadowicz et al., 2016).

The array of SPMS designs and the non-quantitative nature of SPMS mass spectra combined with the wide particle-to-particle mass spectral variation for particles even of similar particle composition, demands flexible data analysis techniques. This is already noted briefly in the paper and the concept will be very familiar to SPMS users. The authors know of no algorithmic methods or associated thresholds utilizing mass spectral information that have been shown to both reliably separate distinct particle types across a large range of chemical compositions and also maintain single particle types within a single cluster. We now emphasize at the end of section 4.1 that this is still an area of active research and discussion in the SPMS field and provide a number of references on the topic. However as stated previously this topic is well beyond the scope of this paper to discuss. A promising potential technique for creating accurate particle types from SPMS datasets is the use of discontinuities in common particle data, such as particle size as discussed by Zelenyuk et al. (2008b), but could also be

extended to values such as total ion intensity. Because this technique is not dependent on specific ion markers, it has the potential to be effective for a broad range of particle types, but is yet to be fully explored. In addition the technique as described by Zelenyuk et al. (2008b) still requires user input to subjectively identify discontinuities. This work by Zelenyuk et al. (2008b) was already briefly referenced at the end of section 4.1, but the discussion has now been expanded slightly to highlight the potential of the "discontinuity technique."

Response to L95: FATES has been tested for compatibility with MATLAB 2016b and is compatible. The text has been changed to reflect this.

Response to L109: The first instance of 'id' has now been changed to 'identifier (ID)'. All further instances of 'id' have been changed to ID.

Response to Section 2.2: A table has now been included to summarize the various run times. While FATES has been updated to be able to import and utilize ALABAMA and TSI-ATOFMS data, we have not included run times beyond study creation. All run times outside of "study creation" will be comparable across SPMS datasets. All SPMS datasets are held in the same FATES data architecture, so data retrieval and analysis is controlled by the size of the data to be retrieved and the FATES data architecture itself. It should be noted that the datasets provided by other laboratories (ALABAMA and TSI-ATOFMS) were only provided to perform proof-of-concept. As such we were only given small amounts of data: 10,000 hit particles from ALABAMA and 68,400 particles for the TSI-ATOFMS. The run times to initiate study creation for these test datasets are included in Table 2 and discussed briefly in Section 2.2. In addition section 2.2 has been reorganized slightly to make it easier to follow.

Reponse to L318: Please see the response to comment 2. In addition, we would like to emphasize that FATES is a toolkit, and is not a technique. The FATES toolkit allows users to organize SPMS datasets in MATLAB in a standardized architecture regardless of specific SPMS instrumentation. This allows datasets to be easily shared

between labs and that continuity can be maintained within labs even as improvements and additional functionalities to individual SPMS datasets are made. FATES is an open-source toolkit, meaning that users have easy access to all code associated with FATES. Therefore, users may modify scripts if needed for their individual lab's purposes but also there is complete transparency for how the data is imported, stored, processed, and analyzed. FATES has been designed to minimize memory demands and computational time required for processing importing, storing, recalling, and analyzing particle and spectral data. Once an SPMS dataset is stored as a FATES study users can apply a virtually unlimited set of analysis techniques utilizing native MATLAB functions. In addition within the FATES toolkit there are a set of visual interactive data analysis tools have also been provided, which can act as a complement to script and algorithmic-based analysis techniques. In summary, FATES is a toolkit intended to enable knowledge discovery from complex SPMS datasets. However, as discussed extensively in the reply to comment #2, laying out "rules" for the wide array of available analyses is well beyond the scope of this paper.

In addition, the visual tools provided in FATES can also be utilized in exploration of datasets consisting of a single particle type. Many papers have been published detailing the effects of laser power (Steele et al., 2005; Wenzel and Prather, 2004), size (Reinard and Johnston, 2008; Zelenyuk et al., 2008a), water content (Neubauer et al., 1997, 1998), and spatial chemical heterogeneity (e.g. Gross et al., 2000; Wenzel and Prather, 2004; Zelenyuk et al., 2008) on laboratory-generated uniform particle populations. In these cases, because the user is not attempting to create distinct particle types, but rather understand the influences of particle and experimental characteristics on the mass spectra generated where algorithmic grouping utilizing mass spectra is largely unnecessary or even inappropriate. In these cases there is no "satisfactory" result, because the goal is not to create or identify particle types, but to more amorphously understand multivariate influences on mass spectral generation.

Response to Section 4.4: As with many aspects of SPMS data analysis, there are

not standard procedures for mass calibration across SPMS designs, and attempts to create standards are somewhat fraught by the mass spectral particle-to-particle variation. While not discussed extensively in the literature, the time of flight of specific m/z will vary depending on the laser fluence experienced by the particle, the location of the particle in the ionization/extraction region of the mass spectrometer, and also the degree to which the particle's chemical components absorb the ionizing radiation. This means that with even careful creation of calibration parameters, there is still often variance in calculated m/z values for what should be identical ion peaks. We are only familiar with the standards used for calibration used in the Kim Prather lab at UCSD, which still result in imperfect calibrations for some particles due to the reasons detailed above. Other laboratories likely have different standards, and it is not unlikely that best practices would vary between instrumental designs. The paper has been modified to gently suggest the methods utilized in Kim Prather's lab at UCSD, but we are hesitant to label such as "standards".

Response to Figures: The annotations on the figures have been enlarged. In addition the resolution of the figures have been greatly increased to allow closer examination.

Ault, A. P., Williams, C. R., White, A. B., Neiman, P. J., Creamean, J. M., Gaston, C. J., Ralph, F. M. and Prather, K. A.: Detection of Asian dust in California orographic precipitation, J. Geophys. Res. Atmos., 116(16), 1–15, doi:10.1029/2010JD015351, 2011.

Bhave, P. V., Allen, J. O., Morrical, B. D., Fergenson, D. P., Cass, G. R. and Prather, K. a.: A Field-Based Approach for Determining ATOFMS Instrument Sensitivities to Ammonium and Nitrate, Environ. Sci. Technol., 36(22), 4868–4879, doi:10.1021/es015823i, 2002.

Cahill, J. F., Fei, H., Cohen, S. M. and Prather, K. a.: Characterization of core–shell MOF particles by depth profiling experiments using on-line single particle mass spectrometry, Analyst, 140(5), 1510–1515, doi:10.1039/C4AN01913J, 2015.

Cai, Y., Zelenyuk, A. and Imre, D.: A High Resolution Study of the Effect of Morphology On the Mass Spectra of Single PSL Particles with Na-containing Layers and Nodules, Aerosol Sci. Technol., 40(12), 1111–1122, doi:10.1080/02786820601001677, 2006.

Creamean, J. M., Suski, K. J., Rosenfeld, D., Cazorla, A., Demott, P. J., Sullivan, R. C., White, A. B., Ralph, F., Minnis, P., Comstock, J. M., Tomlinson, J. M. and Prather, K. A.: Dust and Biological Aerosols from the Sahara and Asia Influence Precipitation in the Western U.S., Science (80-. )., 339(6127), 1572–1578, doi:10.1126/science.1227279, 2013.

Creamean, J. M., Lee, C., Hill, T. C., Ault, A. P., DeMott, P. J., White, A. B., Ralph, F. M. and Prather, K. A.: Chemical properties of insoluble precipitation residue particles, J. Aerosol Sci., 76, 13–27, doi:10.1016/j.jaerosci.2014.05.005, 2014.

Dallosto, M. and Harrison, R.: Chemical characterisation of single airborne particles in Athens (Greece) by ATOFMS, Atmos. Environ., 40(39), 7614–7631, doi:10.1016/j.atmosenv.2006.06.053, 2006. Ge, Z., Wexler, A. S. and Johnston, M. V.: Laser Desorption/Ionization of Single Ultrafine Multicomponent Aerosols, Environ. Sci. Technol., 32(20), 3218–3223, doi:10.1021/es980104y, 1998.

Giorio, C., Tapparo, A., Dall'Osto, M., Harrison, R. M., Beddows, D. C. S., Di Marco, C. and Nemitz, E.: Comparison of three techniques for analysis of data from an Aerosol Time-of-Flight Mass Spectrometer, Atmos. Environ., 61, 316–326, doi:10.1016/j.atmosenv.2012.07.054, 2012.

Gross, D. S., Gälli, M. E., Silva, P. J. and Prather, K. a: Relative sensitivity factors for alkali metal and ammonium cations in single-particle aerosol time-of-flight mass spectra., Anal. Chem., 72(2), 416–22 [online] Available from: http://www.ncbi.nlm.nih.gov/pubmed/10658339, 2000.

Gross, D. S., Atlas, R., Rzeszotarski, J., Turetsky, E., Christensen, J., Benzaid, S., Olson, J., Smith, T., Steinberg, L., Sulman, J., Ritz, A., Anderson, B., Nelson, C.,

Musicant, D., Chen, L., Snyder, D. and

Schauer, J.: Environmental chemistry through intelligent atmospheric data analysis, Environ. Model. Softw., 25(6), 760–769, doi:10.1016/j.envsoft.2009.12.001, 2010.

Hinz, K. and Spengler, B.: Instrumentation, data evaluation and quantification in on-line aerosol mass spectrometry, J. Mass Spectrom., 42, 843–860, doi:10.1002/jms, 2007.

Hinz, K. P., Erdmann, N., Grüning, C. and Spengler, B.: Comparative parallel characterization of particle populations with two mass spectrometric systems LAMPAS 2 and SPASS, Int. J. Mass Spectrom., 258(1–3), 151–166, doi:10.1016/j.ijms.2006.09.008, 2006.

Holecek, J. C., Spencer, M. T. and Prather, K. a.: Analysis of rainwater samples: Comparison of single particle residues with ambient particle chemistry from the northeast Pacific and Indian oceans, J. Geophys. Res. Atmos., 112(22), doi:10.1029/2006JD008269, 2007.

Murphy, D. M., Middlebrook, a. M. and Warshawsky, M.: Cluster Analysis of Data from the Particle Analysis by Laser Mass Spectrometry (PALMS) Instrument, Aerosol Sci. Technol., 37(April 2014), 382–391, doi:10.1080/02786820300971, 2003.

Nash, D. G., Baer, T. and Johnston, M. V.: Aerosol mass spectrometry: An introductory review, Int. J. Mass Spectrom., 258(1–3), 2–12, doi:10.1016/j.ijms.2006.09.017, 2006.

Neubauer, K. R., Johnston, M. V. and Wexler, A. S.: On-line analysis of aqueous aerosols by laser desorption ionization, Int. J. Mass Spectrom. Ion Process., 163(1–2), 29–37, doi:10.1016/S0168-1176(96)04534-X, 1997.

Neubauer, K. R., Johnston, M. V and Wexler, A. S.: Humidity effects on the mass spectra of single aerosol particles, Atmos. Environ., 32(14–15), 2521–2529, doi:10.1016/S1352-2310(98)00005-3, 1998.

Phares, D. J., Rhoads, K. P., Wexler, A. S., Kane, D. B. and Johnston, M. V.: Applica-

tion of the ART-2a Algorithm to Laser Ablation Aerosol Mass Spectrometry of Particle Standards, Anal. Chem., 73(10), 2338–2344, doi:10.1021/ac0015063, 2001.

Pratt, K. A. and Prather, K. A.: Size , and Chemical Composition Measurements of Aged Urban Aerosols, Environ. Sci. Technol., 43(21), 8276–8282, 2009.

Pratt, K. a., DeMott, P. J., French, J. R., Wang, Z., Westphal, D. L., Heymsfield, A. J., Twohy, C. H., Prenni, A. J. and Prather, K. a.: In situ detection of biological particles in cloud ice-crystals, Nat. Geosci., 2(6), 398–401, doi:10.1038/ngeo521, 2009.

Qin, X. and Prather, K. a.: Impact of biomass emissions on particle chemistry during the California Regional Particulate Air Quality Study, Int. J. Mass Spectrom., 258(1–3), 142–150, doi:10.1016/j.ijms.2006.09.004, 2006.

Qin, X., Pratt, K. A., Shields, L. G., Toner, S. M. and Prather, K. A.: Seasonal comparisons of single-particle chemical mixing state in Riverside, CA, Atmos. Environ., 59, 587–596, doi:10.1016/j.atmosenv.2012.05.032, 2012.

Rebotier, T. P. and Prather, K. a: Aerosol time-of-flight mass spectrometry data analysis: a benchmark of clustering algorithms., Anal. Chim. Acta, 585(1), 38–54, doi:10.1016/j.aca.2006.12.009, 2007.

Reinard, M. S. and Johnston, M. V: Ion Formation Mechanism in Laser Desorption Ionization of Individual Nanoparticles., J. Am. Soc. Mass Spectrom., 19(3), 389–399, doi:10.1016/j.jasms.2007.11.017, 2008.

Silva, P. and Prather, K.: Interpretation of mass spectra from organic compounds in aerosol time-of-flight mass spectrometry, Anal. Chem., 72(15), 3553–62 [online] Available from: http://www.ncbi.nlm.nih.gov/pubmed/10952542, 2000.

Silva, P. J., Prather, K., Noble, C. A. and Prather, K. A.: Size and Chemical Characterization of Individual Particles Resulting from Biomass Burning of Local Southern California Species Size and Chemical Characterization of Individual Particles Resulting from Biomass Burning of Local Southern California Species, Environ. Sci. Technol.,

33(18), 3068–3076, doi:10.1021/es980544p, 1999.

Spencer, M. T. and Prather, K. a.: Using ATOFMS to Determine OC/EC Mass Fractions in Particles, Aerosol Sci. Technol., 40(8), 585–594, doi:10.1080/02786820600729138, 2006.

Steele, P. T., Srivastava, A., Pitesky, M. E., Fergenson, D. P., Tobias, H. J., Gard, E. E. and Frank, M.: Desorption / Ionization Fluence Thresholds and Improved Mass Spectral Consistency Measured Using a Flattop Laser Profile in the Bioaerosol Mass Spectrometry of Single Bacillus Endospores, Anal. Chem., 77(22), 7448–7454, 2005.

Sullivan, R. C. and Prather, K. a: Recent Advances in Our Understanding of Atmospheric Chemistry and Climate Made Possible by On-Line Aerosol Analysis Instrumentation Recent Advances in Our Understanding of Atmospheric Chemistry and Climate Made Possible by On-Line Aerosol Analysis Instrum, Anal. Chem., 77(12), 3861–3886, doi:10.1021/ac050716i, 2005.

Wenzel, R. J. and Prather, K. a.: Improvements in ion signal reproducibility obtained using a homogeneous laser beam for on-line laser desorption/ionization of single particles, Rapid Commun. Mass Spectrom., 18(13), 1525–1533, doi:10.1002/rcm.1509, 2004.

Zawadowicz, M. A., Froyd, K. D., Murphy, D. M. and Cziczo, D. J.: Proper identification of primary biological aerosol particles using single particle mass spectrometry, Atmos. Chem. Phys. Discuss., doi:10.5194/acp-2016-1119, 2016.

Zelenyuk, A., Imre, D., Nam, E. J., Han, Y. and Mueller, K.: ClusterSculptor: Software for expert-steered classification of single particle mass spectra, Int. J. Mass Spectrom., 275, 1–10, doi:10.1016/j.ijms.2008.04.033, 2008a.

Zelenyuk, A., Juan, Y., Chen, S., Zaveri, R. a. and Imre, D.: "Depth-profiling" and quantitative characterization of the size, composition, shape, density, and morphology of fine particles with SPLAT, a single-particle mass spectrometer, J. Phys. Chem. A,

112(4), 669–671, doi:10.1021/jp077308y, 2008b.